# Temperature Dependence of Thermal Conductivity of Giant-Scale Supported Monolayer Graphene

**DOI:** 10.3390/nano12162799

**Published:** 2022-08-15

**Authors:** Jing Liu, Pei Li, Shen Xu, Yangsu Xie, Qin Wang, Lei Ma

**Affiliations:** 1College of New Materials and New Energies, Shenzhen Technology University, Shenzhen 518116, China; 2School of Mechanical and Automotive Engineering, Shanghai University of Engineering Science, Shanghai 201620, China; 3College of Chemistry and Environmental Engineering, Shenzhen University, Shenzhen 518055, China

**Keywords:** giant-scale graphene, thermal conductivity, supported by PMMA, compressive strain

## Abstract

Past work has focused on the thermal properties of microscale/nanoscale suspended/supported graphene. However, for the thermal design of graphene-based devices, the thermal properties of giant-scale (~mm) graphene, which reflects the effect of grains, must also be investigated and are critical. In this work, the thermal conductivity variation with temperature of giant-scale chemical vapor decomposition (CVD) graphene supported by poly(methyl methacrylate) (PMMA) is characterized using the differential transient electrothermal technique (diff-TET). Compared to the commonly used optothermal Raman technique, diff-TET employs joule heating as the heating source, a situation under which the temperature difference between optical phonons and acoustic phonons is eased. The thermal conductivity of single-layer graphene (SLG) supported by PMMA was measured as 743 ± 167 W/(m·K) and 287 ± 63 W/(m·K) at 296 K and 125 K, respectively. As temperature decreased from 296 K to 275 K, the thermal conductivity of graphene was decreased by 36.5%, which can be partly explained by compressive strain buildup in graphene due to the thermal expansion mismatch.

## 1. Introduction

The discovery of graphene created explosive growth in research on the various properties of graphene [1,2,3,4]. Due to its high thermal conductivity (*k*) [5], ultrahigh electron mobility [6], and remarkable mechanical strength [7], graphene is regarded to have broad and promising applications including flexible electronics [8,9], photovoltaic devices [10], batteries [11], and so on [12]. With the increasing intense heat of electronics, graphene with super-high *k* has potential applications for the thermal management of microelectronics. A massive amount of work has focused on the experimental and theoretical calculation of the *k* of suspended and supported graphene, and great progress has been achieved [13,14,15,16,17]. The effect of various parameters such as system size [17], defects [18,19,20], substrates [21,22], and grain size [23,24,25] on the *k* of graphene has been deeply studied. Last but not least, since temperature affects the phonon propagation, the *k* of graphene shows strong temperature dependence. It has been reported that the *k* of supported graphene increases with increasing temperature, with the peak *k* appearing around 300 K [14]. When the temperature is lower than 300 K, the phonon-substrate scattering dominates the phonon scattering [14].

Thermal property measurement is critical to the application of graphene in thermal management. Since graphene is extremely thin, the commonly used thermal conductivity measurement techniques of graphene are optothermal Raman spectroscopy [26,27,28] and the micro-bridge method [14]. For optothermal Raman spectroscopy, the accuracy of the measured *k* depends on the absorbed laser power and the temperature coefficient of the Raman shift of the G peak [29]. Li et al. developed the laser-flash optothermal Raman technique, which eases the laser absorption coefficient uncertainty in the measurement of thermal conductivity [29,30]. It is assumed that optical phonons and acoustic phonons are in thermal equilibrium under intense laser irradiation. However, Ruan et al. first reported that the phonon branches were in strong thermal non-equilibrium by employing density functional perturbation theory. The temperature of out-of-plane acoustic phonons (ZA) is 14.8% lower than that of transverse optical phonons (TO) [31]. Under this situation, the *k* of graphene will be overestimated. Wang et al. first detected the lumped temperature difference between optical and acoustic phonons by nanosecond energy transport state-resolved Raman spectroscopy [32]. Seol et al. developed the micro-bridge method to measure the *k* of supported graphene. The micro-bridge method uses joule heating as the heating source. In the experiment, the graphene supported by SiO_2_ acts as the thermal bridge between the heating part and the temperature-sensing part [14]. In principle, the micro-bridge method is feasible, but its measurement accuracy is guaranteed when the thermal resistance of the sample is comparable to that of the resistance thermometer in the experiment setup. In addition, it is technically challenging and time consuming to fabricate the whole measurement device [33].

So far, most thermal conductivity measurements of graphene focus on microscale samples. It is reported that the *k* of suspended graphene is size/geometry dependent. The *k* of graphene has a ~log *L* (*L*: sample length) dependence which was confirmed by experiments (up to 9 μm) [17] and theoretical simulations [34]. Although the *k* of supported graphene is less sensitive to the length size due to substrate coupling [21,35], the *k* of macroscale supported graphene is critical, since the edge-phonon scattering usually leads to a great reduction in *k*. Furthermore, graphene has important applications in the field of flexible electronics, which calls for the thermal properties of graphene supported by flexible materials. To date, seldom work has been conducted on the temperature dependence of the thermal conductivity of millimeter-scale graphene on soft substrates, yet such knowledge is essential to graphene-based device design and optimization.

In this work, the temperature dependence of the thermal conductivity of millimeter-scale SLG supported by poly(methyl methacrylate) (PMMA) is characterized by a differential transient thermoelectrical technique (diff-TET) [36]. The diff-TET employs self-heating as the heating source, which ensures the thermal equilibrium among phonon branches. The sample size is approximately 2–3 mm in width and 1–3 mm in length, which significantly suppresses the effect of size on thermal conductivity. The *k* of PMMA is much lower than that of graphene. Choosing extremely thin (~457 nm) PMMA as the substrate can ensure the contribution of graphene to the overall thermal conductivity is comparable to that of PMMA. Thus, it is feasible to measure the *k* of graphene with high accuracy. The *k* of supported SLG is reported to be 743 ± 167 W/(m·K) at 296 K, which is 23.8% higher than that of micro-scale SiO_2_-supported graphene measured using the micro-bridge method [14]. The *k* of supported SLG is reduced to 287 ± 63 W/(m·K) when the temperature is 125 K, partly due to the strain-induced rips in the sample.

## 2. Sample Preparation and Characterization

### 2.1. Sample Preparation

SLG supported by PMMA characterized in this work was obtained from Jiangsu Xianfeng nanoscale materials company. The samples used in this experiment were fabricated using the chemical vapor deposition (CVD) method. The graphene is grown on a copper (Cu) in a controlled chamber pressure CVD system. At first, a clean Cu foil was annealed at 1077 °C with a H_2_ flow rate of 500 sccm. Then, the H_2_ flow rate and chamber pressure were adjusted to 70 sccm and 108 Torr, respectively. By introducing 0.15 sccm CH_4_ into the chamber, the graphene starts to grow. The copper was etched off after PMMA was coated on the graphene. Finally, graphene supported by PMMA was transferred to a filter substrate. The layer number of graphene was provided in the product technical data.

The as-received graphene was cut into the desired size by scissors. Then, the experimental sample was suspended between two electrodes, as shown in Figure 1a. The effective *k* of these samples will be measured by diff-TET without any further processing. The thickness of PMMA (*δ_p_*) is necessary to determine the *k* of graphene when subtracting the effect of PMMA. Atomic force microscopy (AFM) was used to determine the thickness of PMMA. The supported graphene was transferred to a silicon substrate; then, the thickness of PMMA was determined with AFM. Figure 1b shows the AFM image of the as-received supported graphene. The thickness of PMMA was determined to be 457 nm.

### 2.2. Structure Study Based on Raman Spectroscopy

Even though the layer number of the as-received sample was given in the technical data sheet, the layer number of the as-received graphene on PMMA was verified with confocal Raman spectroscopy (Horiba, LabRam Odyssey). In total, 30 random spots were tested. In the Raman spectroscopy test, a 532 nm laser with ~0.45 mW was focused on the graphene under 50× objective. The integration time was 10 s. Spectra of the as-received sample are shown in Figure 2. No D peak (~1340 cm^−1^) can be observed in the Raman spectra, indicating that the supported graphene has rare D band-related defects. Peaks at 1588.3 cm^−1^ (G band) and 2681.5 cm^−1^ (2D band) can be observed. We used a ratio (*I*_G_/*I*_2D_) of the intensity of the G band to the intensity of 2D band to evaluate the layer number of the as-received sample [37]. It was found that the layer number of each tested spot is 1. Thus, we can verify that the layer number of graphene in the whole sample is 1, and the graphene distributes on the PMMA uniformly. Since the to-be-measured sample was cut from the as-received sample, we can confirm that the graphene layer number of each to-be-measured sample is 1. The PMMA thickness, graphene layer number, length, and width for the four samples are summarized in Table 1.

## 3. Thermal Transport Characterization of Giant-Scale Graphene Supported by PMMA

The thermal diffusivity of graphene supported by PMMA was characterized by diff-TET [36]. As shown in Figure 1a, the sample was suspended between two electrodes. Silver paste was used to eliminate the thermal contact resistance and electric contact resistance between the sample and the electrodes. The whole sample was placed in a vacuum chamber (Janis CCS-100/204N) to eliminate the convective effect on the measurement of the thermal diffusivity. During the experiment, a step current provided by Keithley 6221 was fed through the sample to induce joule heating. The rise of the temperature led to a change in resistance and voltage. The voltage evolution of the sample was recorded by an oscilloscope (Tektronix MDO32). The normalized temperature rise obtained from the experiment is determined as $T^{*}=(V-V_{0})/(V_{1}-V_{0})$. Here, *V*_0_ and *V*_1_ are the initial voltage and steady-state voltage over the sample, respectively. The environmental temperature was controlled by the temperature controller which ranges from 7 K to 300 K. Since most of the samples were broken at 125 K with a sudden jump resistance to a very high value, we did the experiments from room temperature (RT) down to 125 K in 25 K steps. During the joule heating, the thermal transport in the sample can be regarded as one-dimensional, thus the energy governing equation can be expressed as:(1)$$\frac{\partial (\rho c_{p}T)}{\partial t}=k\frac{\partial^{2}T}{\partial x^{2}}+\dot{q}$$
where *ρ*, *c_p_*, and *k* are the density, specific heat capacity of the sample, respectively. $\dot{q}$ is the heating power per unit volume, which equals to $I^{2}R_{s}/AL$. Here, *A* and *L* are the cross-sectional area and length of the sample, respectively. *I* and *R_s_* are the fed-in current and the sample resistance, respectively. Since the two electrodes are much larger than the sample, the temperature of the two electrodes can be regarded as environmental temperature. Thus, the boundary conditions can be depicted as: *T*(*x* = 0, *x* = *L*) = *T*_0_ (*T*_0_: room temperature). The theoretical normalized temperature rise is defined as: $T^{*}(t)=[T(t)-T_{0}]/[T(t\to \infty )-T_{0}]$. The theoretical solution to Equation (1) is solved as:(2)$$T^{*}=\frac{48}{\pi^{4}}\sum_{m=1}^{\infty }\frac{1-\exp[-{\left(2m-1\right)}^{4}\pi^{2}\alpha_{eff}t/L^{2}]}{{\left(2m-1\right)}^{4}}$$

A Matlab program created based on Equation (2) was used to fit the normalized temperature rise obtained by the experiments. The trial value of *α_eff_* which gives the best fit of the experiment data was taken as the sample’s effective thermal diffusivity. Figure 3b shows the normalized voltage evolutions and the fitting curves for S1 at 275, 200, and 125 K. Magnificent fitting was obtained. The TET technique has been rigorously proven to be a quick and effective method to measure the thermal diffusivity of various conductive and non-conductive micro/nanoscale samples. More details can be found in references [13,36].

The effective thermal diffusivity (*α_eff_*) includes the radiation effect (*α_rad_*) which can be expressed as:(3)$$\alpha_{rad}=8\varepsilon k_{B}T^{3}L^{2}/(\pi^{2}\delta \rho c_{p})$$
where *ε*, *k_B_*, and *δ* are emissivity, the Stefan–Boltzmann constant, and thickness of the sample, respectively. The emissivity of the sample is taken from [36] as 0.08. After obtaining the real thermal diffusivity (*α_real_*), the effective thermal conductivity (*k_eff_*) can be obtained as *k_eff_* = *α_real_*·(*ρc_p_*)*_p_*. The volumetric heat capacity of PMMA was used for the whole sample with high accuracy due to the extremely low mass proportion of graphene. In part 2, the Raman spectroscopy study of graphene showed that the graphene is distributed on the PMMA uniformly and all the graphene in samples is found to be a single layer. The relationship between effective thermal conductivity and thermal conductivity of PMMA and graphene can be depicted as:(4)$$k_{eff}=\frac{k_{g}\delta_{g}+k_{p}\delta_{p}}{\delta_{g}+\delta_{p}}$$

Here, *δ_p_* and *δ_g_* are the thickness of PMMA and graphene, and are taken as 457 nm and 0.335 nm, respectively. *k_p_* and *k_g_* are thre thermal conductivities of PMMA and graphene, respectively. The temperature-dependent volumetric heat capacity and thermal conductivity of PMMA are taken from references [38,39]. The test methods of the heat capacity and *k* of PMMA are the vacuum calorimeter and the 3ω method, respectively [38,39]. The thermal conductivity of SLG supported by PMMA is obtained from the average graphene thermal conductivities of the four samples. Two critical points should be explained here. In thermal characterization, the graphene is first heated by the electrical current, then the thermal energy is transferred to the PMMA. Work by Liu et al., has proven that graphene and PMMA reached thermal equilibrium in the cross-sectional direction during thermal characterization [36]. Additionally, the interface thermal resistance (~10^−2^ K/W) between graphene and PMMA is neglectable in comparison to the thermal resistance (~10^6^ K/W) of PMMA. More details can be found in [36].

## 4. Results and Discussion

### 4.1. Abnormal Temperature Coefficient of Resistance for Graphene Supported by PMMA

The normalized resistance (*R** = *R*/*R*_0_, *R*_0_: resistance at RT) variation against the temperatures of S1, S2, S3, and S4 is presented in Figure 4a. The resistances of the four samples jump to very high values at 100 K, which indicates severe fractures in the sample. Thus, the normalized resistance is shown to be between RT and 125 K. It has been reported that the resistivity (*ρ_e_*) of graphene is linearly proportional to *T*, as temperature is higher than the Bloch–Gruneisen temperature (*Θ_BG_*) of graphene and is proportional to *T*^4^ in the opposite case (*T < Θ_BG_*) [40]. However, the *R** of the four samples decreases slightly as temperature decreases, then increases as temperature is decreased from RT to 125 K. The turning point is in the range of 250–275 K. The *R** of S2 and S4 is increased by 35% and 23%, while the *R** of S1 and S3 is decreased by 0.012% and 0.0075% as temperature changes from RT to 125 K. To obtain more details on the resistance of supported graphene, the temperature dependence of *R** of S5 was measured every 10–25 K for three rounds. Figure 4b shows the normalized resistance of three rounds for S5. As temperature decreases from RT to 150 K, the *R** trends of S5 are similar to that of S1 and S3. As temperature continues to decrease from 150 K, the *R** of S5_r1 and S5_r2 first decreases and then increases with temperature, while the *R** of S5_r3 continues to increase. Hinnefeld et al. studied the resistance of graphene supported by PDMS under strain and it was found that the tensile strain in the supported graphene can lead to rips in the graphene. These rips cause the resistance of the supported graphene to increase significantly with increased strain [41]. As temperature decreases, the graphene will expand and PMMA will contract. However, as the graphene is adhered to PMMA by van der Waals forces, the graphene will contract. The Raman spectra study of the experimental sample in the following part also confirms that the compressive strain builds in graphene as the temperature decreases from RT to 125.

The temperature coefficient of resistance (TCR) of graphene supported by PMMA is further discussed based on the data of S5_r3. The TCR of S5_r3 is shown in Figure 4c. The inset gives a close up to the TCR as temperature is in the range of 7–180 K. Apart from the intrinsic TCR of relaxed graphene, the overall TCR of the sample is affected by the thermal expansion of PMMA and graphene. The overall TCR of the whole sample can be described by the following equation: $TCR=TCR_{i}+(\beta_{p}-\beta_{g})\gamma$. Here, $TCR_{i}$ and *γ* are the intrinsic TCR of relaxed graphene and a positive constant, respectively. *β_p_* and *β_g_* are the thermal expansion coefficients (*β*) of PMMA and graphene, respectively. Figure 4d shows the TEC of graphene [40] and PMMA [39]. As graphene and PMMA are held together by van der Waals forces, the compressive strain/stress will build in graphene. As shown in Figure 4c, *β_p_* decreases with decreasing temperature and remains positive when the temperature is above 0 K. *β_g_* remains negative when temperature is lower than RT. In the entire temperature range, $TCR_{i}$ remains negative [42,43]. When the temperature is close to RT, *β_p_* is relatively large, which helps maintain a positive TCR in supported graphene. As temperature is decreased, the difference between *β_p_* and *β_g_* becomes smaller in comparison to the effect of $TCR_{i}$, causing the TCR of supported graphene to be negative.

### 4.2. Thermal Properties of SLG Supported by PMMA

The real thermal diffusivity without the effect of radiation of four samples is shown in Figure 5a. As temperature is decreased from RT, the *α_real_* of S1, S2 and S4 decreases first, then the *α_real_* of S2 remains constant while the *α_real_* of S1 and S4 increases slightly. The change in the *α_real_* of S3 is relatively small when compared to the other three samples. The *k* of SLG supported by PMMA is presented in Figure 5b. For comparison, the *k* of SLG supported by SiO_2_ [14,27] and bilayer graphene with PMMA residue [42] are shown in Figure 5b. At RT, the *k* of supported SLG is 743 ± 167 W/(m·K), which is smaller than that of SiO_2_-supported SLG (840 W/(m·K)) [27]. In [27], the *k* of supported graphene is measured by a comprehensive Raman optothermal method which eliminated the laser absorption uncertainty [27]. However, this work did not take the thermal non-equilibrium between different phonon branches into account. In this situation, the *k* of supported graphene is overestimated. The *k* in this work is larger than two literature values (~600 W/(m·K) for SiO_2_-supported SLG [14] and 560 W/(m·K) for bilayer graphene with PMMA [42]). The differences in the *k* of our work and above two differences may be attributed to the different interface coupling strength.

As temperature decreases from RT to 275 K, the *k* of supported SLG first decreases by 36.5%, which is much larger than the *k* reduction in [14,27]. As discussed in Section 4.1, there is compressive strain that builds in the graphene as temperature decreases from RT. There is limited work about the strain effect on the *k* of graphene supported by a flexible substrate. As such, we turn our attention to the effect on the *k* of suspended graphene. As reported in the literature, compressive strain in suspended graphene suppresses thermal conductivity [43]. This is attributed to the fluctuant structure created in graphene under compressive strain, under which the phonons can be scattered significantly more often [43,44]. With −0.05% strain in suspended graphene, it has been reported that the *k* can be reduced by 26% [43]. We speculate that the compressive strain induced by the TEC mismatch between graphene and PMMA results in a large *k* reduction when temperature decreases from RT to 275 K. Thus, Raman spectroscopy is used to investigate the G peak under different temperatures. The graphene sample suspended between two electrodes was placed in a cryogenic cell where the temperature can be controlled between 77 and 300 K. A confocal Raman is used to collect the Raman spectra of graphene under different temperatures.

Figure 6a shows the frequency shift of the G peak (Δ*ω_G_*) of SLG on PMMA as a function of temperature. The G peak blueshifts as temperature goes down from RT. The temperature dependent frequency shift of the G peak is attributed to three parts: thermal expansion of the lattice ($\Delta \omega_{G}^{E}$); anharmonic effect ($\Delta \omega_{G}^{A}$), changing the phonon energy; and compressive strain ($\Delta \omega_{G}^{S}$), induced by the TEC mismatch between graphene and PMMA [45]. Thus, Δ*ω_G_* can be given by [45]:(5)$$\Delta \omega_{G}(T)=\Delta \omega_{G}^{E}(T)+\Delta \omega_{G}^{A}(T)+\Delta \omega_{G}^{S}(T)$$

The G-band frequency shift due to the strain induced by the TEC mismatch can be expressed as [45]:(6)$$\Delta \omega_{G}^{S}(T)=\beta \varepsilon (T).$$
where *β* is the biaxial strain coefficient of the G band. The biaxial strain coefficient of the G band has been determined to be around −70 cm^−1^/% at RT [46]. Since the temperature dependence of the G-band frequency is more reliable than the TEC of graphene and PMMA, the G-band frequency shift due to temperature variation ($\Delta \omega_{G}^{E}$ + $\Delta \omega_{G}^{A}$) obtained from theoretical calculations [47] are subtracted from Δ*ω_G_* to obtain $\Delta \omega_{G}^{S}(T)$. Strain buildup in graphene is calculated according to Equation (6). Figure 6b shows the temperature dependence of the strain buildup in graphene between graphene and PMMA. At 275 K, the strain is −0.047%, which indicates that compressive strain was built in graphene [46]. Li et al. reported that the *k* of suspended graphene can be reduced by 26% with −0.05% strain in graphene [43]. However, whether the −0.047% strain is sufficient to explain the large *k* reduction at 275 K is still doubtful. It also has been reported that the *k* of graphene nanoribbons are insensitive to compressive strain [48]. For graphene supported by flexible substrate, the effect of compressive strain on *k* still needs more theoretical calculations. As temperature continues decreasing from 275 K, the variation of *k* is not obvious, especially in the range of 150–225 K. We interpreted this trend from the structure of the sample. The SEM figure of S3 after cryogenic test is obtained and shown in Figure 7. A lot of micro-scale rips are observed. We speculate that there are micro-rips in the graphene when temperature is lower than 225 K. The rips suppress the *k* of graphene as compared to the *k* of supported graphene without any rips [14,42]. At lower temperatures, the effect of rips on thermal conductivity dominates, which keeps the changes in the *k* of graphene small.

## 5. Conclusions

In summary, we first reported the resistance–temperature relationship for supported graphene since this relationship is critical for explaining the behavior of graphene in thermal characterization. The TCR of our sample reduced from a positive value at RT to a negative value at low temperatures, while the free-standing graphene has a negative TCR throughout the entire temperature range. The abnormal *R–T* relationship is due to the TEC mismatch between graphene and PMMA and rips in graphene appearing at lower temperature. Using the diff-TET technique, the temperature dependence of the thermal conductivity of graphene supported by a flexible substrate is obtained. Through monitoring the blueshift of the Raman peak, compressive strain is confirmed to be built up in the graphene. The large reduction in *k* as temperature decreases from RT to 275 K is partly due to the compressive strain built up in graphene. This work can provide some guidance on the design of graphene-based electrothermal devices.

## Figures and Tables

**Figure 1 nanomaterials-12-02799-f001:**
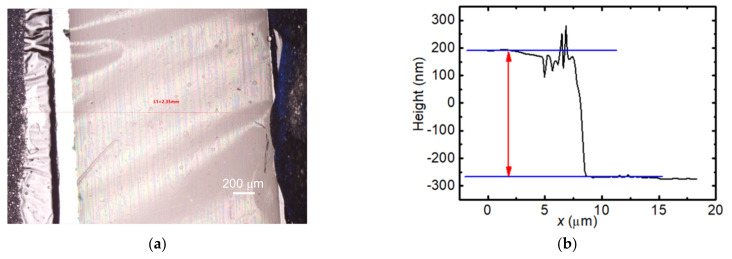
(**a**) Topology of S3 under microscope; (**b**) AFM image of the as-received SLG supported by PMMA.

**Figure 2 nanomaterials-12-02799-f002:**
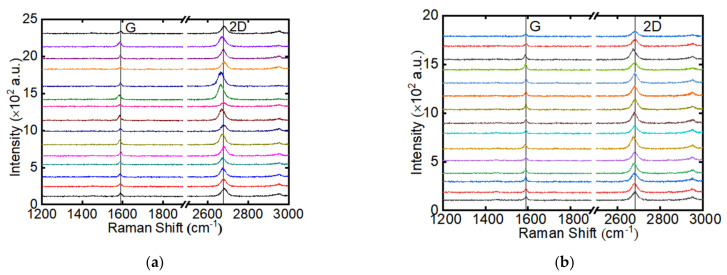
(**a**,**b**) Raman spectra of as-received sample.

**Figure 3 nanomaterials-12-02799-f003:**
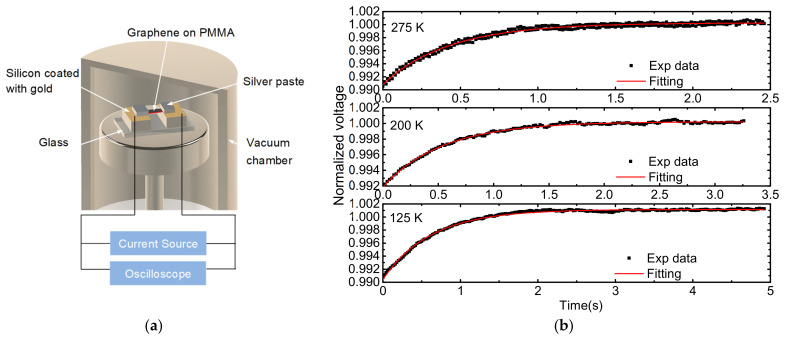
(**a**) Schematic setup of the TET measurement (not to scale); (**b**) Normalized voltage evolution and the fitting curve for S1 at 275, 200, and 125 K.

**Figure 4 nanomaterials-12-02799-f004:**
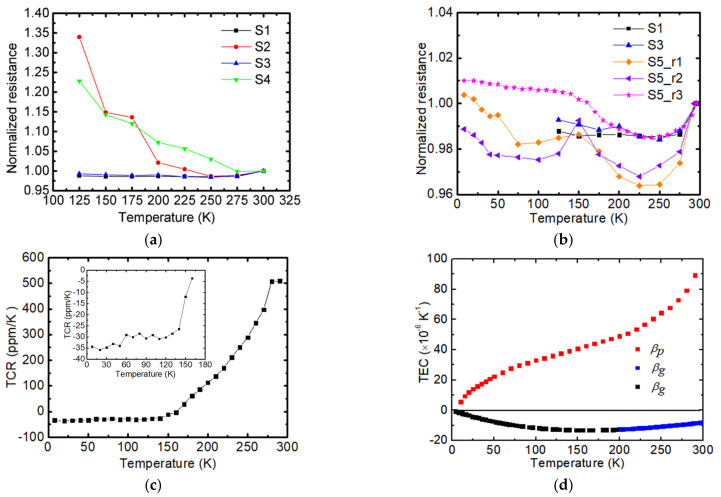
(**a**) Normalized resistance of S1, S2, S3, and S4. (**b**) Normalized resistance of S1, S3, S5_r1, S5_r2. and S5_r3. (**c**) Temperature coefficient of resistance of S5. The TCR was obtained based on the data of S5_r3. (**d**) Thermal expansion coefficient of PMMA (Reprinted/adapted with permission from Ref. [39]. Copyright 2013, Elsevier) and suspended graphene (Reprinted/adapted with permission from Ref. [40]. Copyright 2011, American Chemical Society). The data in blue square are obtained through experiment while the data shown in black square are estimated values.

**Figure 5 nanomaterials-12-02799-f005:**
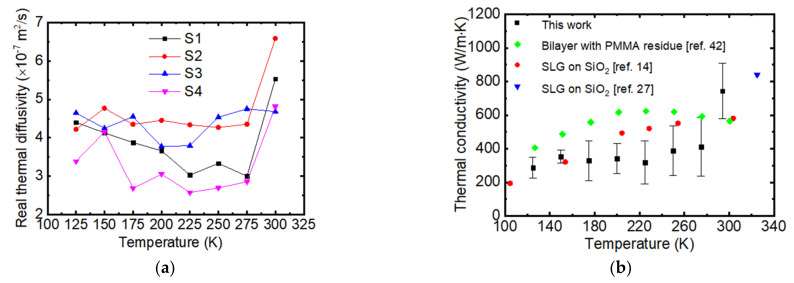
(**a**) Real thermal diffusivity without radiation effect for S1, S2, S3, and S4; (**b**) Thermal conductivity variation against temperature of SLG supported by PMMA. For comparison, the thermal conductivities of SLG supported by SiO_2_ (Reprinted/adapted with permission from Ref. [14]. Copyright 2010, The American Association for the Advancement of Science) and bilayer graphene with PMMA residue (Reprinted/adapted with permission from Ref. [42]. Copyright 2011, American Chemical Society).

**Figure 6 nanomaterials-12-02799-f006:**
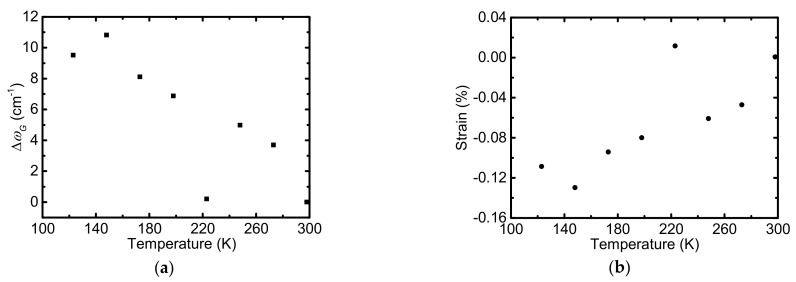
(**a**) Frequency shift of the G band of SLG supported by PMMA. (**b**) Temperature dependence of the strain buildup in graphene due to the TEC mismatch between graphene and PMMA.

**Figure 7 nanomaterials-12-02799-f007:**
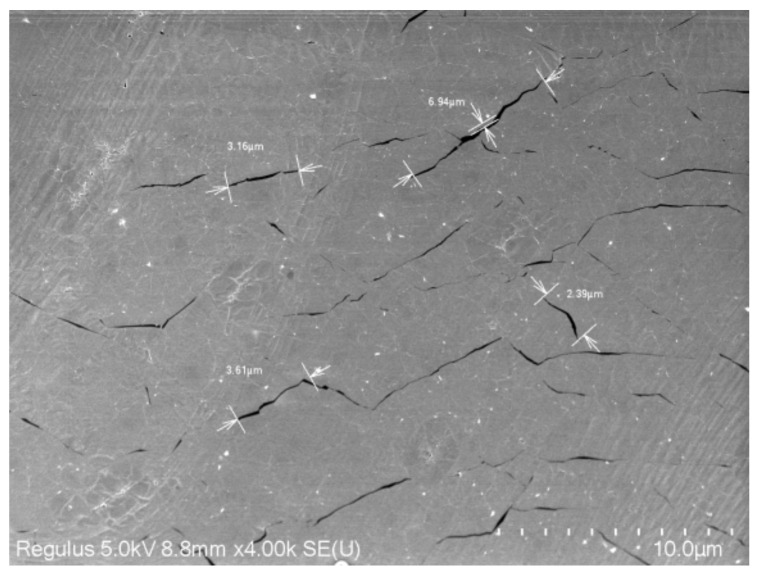
SEM figure of S3 after the cryogenic test. Many micro-scale rips can be observed.

**Table 1 nanomaterials-12-02799-t001:** PMMA thickness, graphene layer number, length, and width of S1, S2, S3, S4, and S5.

Sample	S1	S2	S3	S4	S5
PMMA thickness(nm)	457	457	457	457	457
Graphene layer number	1	1	1	1	1
Length (mm)	1.73	1.53	2.35	2.25	2.09
Width (mm)	1.92	1.85	3.20	2.80	0.74

## Data Availability

The data presented in this study are available on request from the corresponding author.

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
