# Peer review of "Temperature Dependence of Thermal Conductivity of Giant-Scale Supported Monolayer Graphene"

_nanomaterials, 2022, doi:10.3390/nano12162799_

Round 1

Reviewer 1 Report

This paper investigated the thermal property of giant scale graphene. The thermal conductivity variation with temperature was investigated. I have the following questions.

Q1. How much is the fed-in current? Does the current value has a big influence on the results?

Q2. Why did the author choose PMMA? How did it influence the radiation effect? Is it possible to reveal the influence of PMMA if PMMA is coated on both top and bottom surfaces of graphene?

Q3. The sample size is small. How did the authors apply silver paste?

Q4. In line 125, is it Fig. 3(a)?

Reviewer 2 Report

Thanks for the trust of the Editor. I have reviewed the manuscript very carefully. I think the authors’ work on measuring the thermal conductivity of giant-scale supported monolayer graphene is meaningful and interesting. Graphene, as a highly conductive and thermally conductive two-dimensional material, especially the accurate measurement of the thermal conductivity of large-scale supported graphene, is beneficial for efficient thermal management in flexible electronics. However, the manuscript has some problems and is not explicit and rigorous enough. I recommend major revisions. Regarding the manuscript’s content, some suggestions for corrections are listed below.

(1) The effects of size, defect, substrates, and grain size on the k of graphene have been extensively studied. Similarly, as another important parameter, the temperature may impact phonon propagation in graphene. But the corresponding principle in the temperature dependence of k of graphene is lacking in the Introduction. Please supplement this part.

(2) In line 55, “The micro-bridge method is feasible in principle, but its measurement accuracy is guaranteed when the sample has proper thermal resistance” this passage is awkward to read. Please check carefully.

(3) The authors use Raman spectroscopy to determine the layer number of graphene, why do 30 random spots need to be tested, and what is the difference between Fig.2 (a) and (b)?

(4) In line 171, the heat capacity and k of PMMA are obtained from the literature, and the authors should clarify the test method of the literature because different test methods will impact the polymer's thermal properties.

(5) In line 189, “However, R* of the four samples …… from RT to 125K”, the description of the passage is inconsistent with Fig.4 (a). Please check carefully.

(6) For sample S5, I didn’t find a description of it (length and width). Please check carefully.

(7) The diff-TET method is used in the manuscript to measure the k. The electric current initially heats the graphene, but this process involves phonon propagation and electron migration. For heat transfer in conductive materials, the role of electrons cannot be ignored, and the contribution is often more significant than that of phonons. Please explain why the manuscript focuses on phonons.

(8) In line 285, “The SEM figure of S3 after cryogenic test is obtained.”, I didn’t find the corresponding image in the manuscript. Please check carefully.

(9) Please follow the Nanomaterials publication requirements to unify the format of the references. 

Round 2

Reviewer 2 Report

I am glad that the author understood my concerns, and made revisions and explanations to the manuscript, and it is recommended to accept it.